# Seroepidemiology of SARS-CoV-2 in healthcare personnel working at the largest tertiary COVID-19 referral hospitals in Mexico City

Vanessa Dávila-Conn[1ᵒ], Maribel Soto-Nava[1ᵒ], Yanink N. Caro-Vega[2ᵒ], Héctor E. Paz-Juárez[1], Pedro García-Esparza[1], Daniela Tapia-Trejo[1], Marissa Pérez-García[1], Pablo F. Belaunzarán-Zamudio[3], Gustavo Reyes-Terán[4], Juan G. Sierra-Madero[2], Arturo Galindo-Fraga[2], Santiago Ávila-Ríos[1]*

**1** Centre for Research in Infectious Diseases, National Institute of Respiratory Diseases, Mexico City, Mexico, **2** Instituto Nacional de Ciencias Médicas y Nutrición Salvador Zubirán, Mexico City, Mexico, **3** Independent investigator, Bethesda, MD, United States of America, **4** Institutos Nacionales de Salud y Hospitales de Alta Especialidad, Secretaría de Salud de México, Mexico City, Mexico

ᵒ These authors contributed equally to this work.

* santiago.avila@cieni.org.mx

## Abstract

### Introduction

We performed a longitudinal SARS-CoV-2 seroepidemiological study in healthcare personnel of the two largest tertiary COVID-19 referral hospitals in Mexico City.

### Methods

All healthcare personnel, including staff physicians, physicians in training, nurses, laboratory technicians, researchers, students, housekeeping, maintenance, security, and administrative staff were invited to voluntarily participate, after written informed consent. Participants answered a computer-assisted self-administered interview and donated blood samples for antibody testing every three weeks from October 2020 to June 2021.

### Results

A total of 883 participants (out of 3639 registered employees) contributed with at least one blood sample. The median age was 36 years (interquartile range: 28–46) and 70% were women. The most common occupations were nurse (28%), physician (24%), and administrative staff (22%). Two hundred and ninety participants (32.8%) had a positive-test result in any of the visits, yielding an overall adjusted prevalence of 33.5% for the whole study-period. Two hundred and thirty-five positive tests were identified at the baseline visit (prevalent cases), the remaining 55 positive tests were incident cases. Prevalent cases showed associations with both occupational (institution 2 vs. 1: adjusted odds ratio [aOR] = 2.24, 95% confidence interval [CI]: 1.54–3.25; laboratory technician vs. physician: aOR = 4.38, 95% CI: 1.75–10.93) and community (municipality of residence Xochimilco vs. Tlalpan: aOR =

**Data Availability Statement:** The de-identified dataset used for analysis is available as part of the supporting information.

**Funding:** This work was supported by Consejo Nacional de Ciencia y Tecnología (CONACyT) (Fondo FORDECYT-PRONACES) and the Mexican Government (Programa Presupuestal P016; Anexo 13 del Decreto del Presupuesto de Egresos de la Federación) to SAR. Funders had no role in study design, data collection, analysis or interpretation, writing of the report and decision to submit for publication.

**Competing interests:** Authors report no conflicts of interest.

2.03, 95% CI: 1.09–3.79) risk-factors. The incidence rate was 3.0 cases per 100 person-months. Incident cases were associated with community-acquired risk, due to contact with suspect/confirmed COVID-19 cases (HR = 2.45, 95% CI: 1.21–5.00).

## Conclusions

We observed that between October 2020 and June 2021, healthcare workers of the two largest tertiary COVID-19 referral centers in Mexico City had similar level of exposure to SARS-CoV-2 than the general population. Most variables associated with exposure in this setting pointed toward community rather than occupational risk. Our observations are consistent with successful occupational medicine programs for SARS-CoV-2 infection control in the participating institutions but suggest the need to strengthen mitigation strategies in the community.

## Introduction

Severe Acute Respiratory Syndrome Coronavirus 2 (SARS-CoV-2) disease (COVID-19) was first confirmed in Mexico in late February 2020, reaching over 2 million confirmed cases and 200,000 deaths one year later [1]. To face the COVID-19 epidemic, official contingency measures were established in Mexico on March 23rd 2020. These included a non-compulsory stay-at-home policy, protection of highly vulnerable groups, and designation and conditioning of tertiary level institutions as referral centers for COVID-19 care exclusively, to increase hospitalization capacity [2, 3]. These included the National Institute of Respiratory Diseases (INER) and National Institute of Medical Sciences and Nutrition (INCMNZS), two of the largest COVID-19 response centers in Mexico City, where more than a quarter of the total accumulated COVID-19 cases in Mexico have occurred [1]. These institutions are characterized by highly qualified personnel, well-equipped facilities, and better access to medical supplies; both are part of the tertiary healthcare institution network in Mexico, where highly specialized health care and basic, clinical and epidemiological research take place [4].

Before the arrival of the omicron variant, three epidemiological waves of COVID-19 were observed in Mexico, with peaks in July 2020, January 2021, August 2021, and the currently ongoing fourth wave caused by the omicron variant. We enrolled participants during the second wave, which in Mexico City reached an estimated cumulative incidence of 434 cases per 100,000-person, reaching a maximum of 7371 cases per day (and over 20,000 cases per day at the country level) [1, 5]. Preliminary and recently published results of nationally representative seroepidemiological surveys in Mexico estimated that over one third of the general population had been exposed to SARS-CoV-2 in Mexico by December 2020 [6, 7]. While prevalence depends on the moment of estimation, as well as on duration of immunity, it is expected to increase after each epidemiological wave.

Overall, health care workers are expected to have increased risk of exposure to SARS-CoV-2 associated with frequent contact with both asymptomatic and hospitalized COVID-19 patients or with handling biological specimens. While many studies around the world have reported higher prevalence of SARS-CoV-2 infection in healthcare personnel, compared to the general population [8, 9], others report similar prevalence in both groups [10]. The risk of exposure for healthcare personnel may be determined by occupational factors, such as availability of personal protective equipment (PPE), infrastructure, workplace setting, access to

training; in addition to the baseline risk of community-acquired infection. Reported levels of exposure to SARS-CoV-2 in healthcare workers varies widely in different settings and around the world, with studies reporting from low seroprevalence [11–28], to studies reporting seroprevalence over 10% [29–40]. This variation may be additionally influenced by the prevention and mitigation measures established in different countries, the prevalence of chronic diseases in the population, individual risk behaviors, the local stage of the epidemic at the time of the survey, the frequency of screening, and the vaccination coverage in key populations.

As of June 2021, there were nearly 240,000 COVID-19 cases reported in healthcare workers in Mexico, from a total of over 2,240,000 cases nationwide. Among this group, Mexico City had the largest number of cases and deaths, with the highest case fatality rate at the beginning of the epidemic (nearly 6%) with reductions thereof (below 2% by June 2021 compared to 5% in the general population) [5]. This reduction can be attributed to the uptake of COVID-19 vaccines among healthcare workers of COVID-19 referral centers, which were prioritized to start vaccinations in late December 2020.

Seroepidemiological studies can contribute with knowledge on the level of exposure to SARS-CoV-2 and the burden of disease in different populations and epidemiological contexts, estimating the total number of exposure cases, independently of clinical presentation. They can also provide information on the spread and specific transmission risks associated with a pathogen. In the case of healthcare workers, this knowledge can help to better understand occupational hazards, assess the effectiveness of prevention measures, and inform prevention policies. The level of exposure to SARS-CoV-2 in healthcare personnel in Mexico has not been reported. This information could help to identify work-associated risks of infection, which in turn can be used to design more effective occupational medicine programs and optimize vaccination strategies.

Here, we present results of a longitudinal seroepidemiological study in healthcare personnel working at the two largest tertiary referral hospitals in Mexico City functioning as COVID-19-only facilities since the beginning of the pandemic. We aimed to assess the overall prevalence and incidence of SARS-CoV-2 infection in the local context for all occupational risk categories (from physicians and nurses to administrative staff), and analyze specific community- and work-associated exposure hazards.

## Materials and methods

### Study design and settings

This was an observational, prospective, longitudinal, cohort study among healthcare personnel working at the two largest, tertiary care, National Ministry of Health referral hospitals for COVID-19 in Mexico City: the National Institute of Respiratory Diseases (INER) and the National Institute of Medical Sciences and Nutrition Salvador Zubirán (INCMNSZ). These institutions were converted into COVID-19-only referral facilities for severely-ill patients at the beginning of the sanitary crisis in the city. Participants were enrolled between October 2020 and June 2021, and followed-up for up to five visits, separated at least 21 days one from each other. In each visit, we collected blood samples for SARS-CoV-2 antibody testing and applied structured questionnaires to collect information of sociodemographic characteristics, and occupational and community risk exposure.

### Ethics statement

The study was reviewed and approved by the Institutional Review Boards of both participating institutions: The National Institute of Medical Sciences and Nutrition (registry: CONBIOÉTICA-09CEI-011-20160627), project code 3432, and the National Institute of Respiratory

Diseases (registry: CONBIOÉTICA-09CEI-003-20160427), project code C35-20. Participants went through an informed consent process both at registration to the study in the electronic portal and in a written form when attending their first appointment for blood draw. Data was stored in a local server, assuring confidentiality of participants' information. Results from antibody tests were uploaded to each participant's account in the portal, for personal consultation. Only key research personnel and IT staff had access to the data and databases used for the analyses were previously de-identified. Each participant had access to a portal containing their individually scheduled appointments and the results from serological tests. Data privacy and access policies were explained to and accepted by the participants during the informed consent process and before answering the computer-assisted self-administered interview (CASI). The study was conducted according to the principles of the Declaration of Helsinki.

## Study participants

We invited all healthcare personnel of the two institutions to voluntarily participate in the study. We included staff physicians and physicians in training (residents and fellows), nurses, laboratory technicians, researchers, graduate biomedical research students, and housekeeping, maintenance, laundry, security, kitchen, and administrative staff. All occupations were included, given the hypothesized overall increased risk of exposure in the work environment due to the concentration of severely ill persons in the participating institutions, as well as the possibility of frequent contact of co-workers with different risk of exposure to SARS-CoV-2 according to their activities.

Participants were invited to enroll through different institutional communication channels, including advertisements in institutional web pages, official social media accounts (Facebook, Twitter), email lists, conferences, posters, and flyers. Participants were asked to register in an electronic portal designed specifically for the project, answer a computer-assisted self-administered interview (CASI) on occupational and community exposure to SARS-CoV-2 infection, and schedule an appointment to donate a blood sample for antibody testing.

To minimize attrition, we sent email reminders to schedule follow-up appointments ideally every 21 days (plus/minus one week) until a maximum of five follow-up visits were completed.

## Data collection

We collected general sociodemographic information, data on community and occupational exposure, as well as history of symptoms since the beginning of the epidemic upon registration for the study, using a baseline CASI. Data was stored in a local secured server. From the second visit onwards, we applied similar CASI for the occurrence of exposure variables and symptoms in the previous 15 days. General characteristics collected included sex, age, occupation, institution, place of residence, and contact information. For occupational and community exposure, we collected data on access to PPE, contact with COVID-19 patients, compliance with basic preventive measures. The latter included physical distancing, hand washing and use of face masks, which were recorded as categorical variables (always, generally, sometimes, and never). In addition, handling of biological samples, contact with any known or suspect COVID-19 case, and use of public transport to commute to work, were recorded as dichotomous variables (yes, no). The presence or absence of symptoms was also recorded in a dichotomous manner (see S1 Appendix).

## Antibody tests

We measured total anti-SARS-CoV-2 nucleocapsid (N) protein antibodies using an electrochemiluminescence-based commercial assay from EDTA-anticoagulated plasma, as

recommended by the manufacturer (Elecsys Anti-SARS-CoV-2, Roche, Basel, Switzerland), on a Cobas e411 instrument. Results were expressed as cut-off index (COI) values. A COI $\geq 1$ was considered reactive. The reported test sensitivity is 97.92% and specificity 99.95% [41].

## Statistical analyses

We classified participants in three groups based on the serology test results: prevalent cases (participants with positive antibody tests at the first visit), incident cases (participants with a positive test result after having at least the first negative test result) and non-cases (participants that remained seronegative during the study period). Seroprevalence was estimated as the percentage of individuals with a positive antibody test at any time during the study period of the total of enrolled participants. We adjusted prevalence to test performance with the formula:

$$Adjusted\ Prevalence = \frac{Crude\ Prevalence + Specificity - 1}{Sensitivity + Specificity - 1}$$

We estimated the incidence rate excluding the prevalent cases. We counted the number of new positive cases and divided them over the sum of person-time in follow up. The person-time was calculated as follows: incident cases contributed with the time elapsed from the date of their first sample to the date of their first positive sample, and non-cases contributed with their complete follow-up time from their first to their last sample date. Participants with only one follow-up visit contributed with one person-day. We expressed the measure of incidence in cases per 100 person-months of observation. We additionally used a Kaplan-Meier curve to show incidence in the cohort. We report the total number of serological tests and the number of positive tests per week across the study period.

General characteristics, presence of symptoms, compliance with basic preventive measures and occupational hazard factors were analyzed using medians and interquartile ranges or absolute counts and percentages, as appropriate. We constructed univariate and multivariate logistic regression models to explore possible associations between these variables and SARS-CoV-2 exposure for prevalent cases, from answers provided at the baseline questionnaire. To study factors associated with incident cases we used a Cox regression model, stratified by institution, based on the data collected in the questionnaire applied at the visits of the first positive test. Multivariable adjustment included all variables showing significant bivariate associations, and occupational hazard variables, age, and sex, which were included in the model *a priori*.

All analyses were performed using STATA v16 and R version 1.2.5019.

## Results

### Study population

A total of 1,129 individuals registered as volunteers in the electronic database between October 2020 and June 2021. Of these, 883 (78%) turned up to at least one appointment for blood sample collection. The median age of participants was 36 years (interquartile range, IQR: 28–46) and the majority (70%) were women. The most common occupations were nurse (28%), physician (including staff, residents, and fellows) (24%) and administrative staff (22%). Regarding occupational hazards, 37% reported handling biological specimens from patients with COVID-19, and 42% reported frequent contact with patients with COVID-19. Compliance with preventive measures at work was high: 92% reported always or generally using PPE, 99% reported always or generally wearing face masks, and 100% reported frequent hand washing (Table 1). Interestingly, 54% reported having had recent contact with any person with COVID-19 either at work or in the community and 48% used public transport to commute to work.

**Table 1. Baseline characteristics of healthcare personal of two of the largest COVID-19 referral hospitals in Mexico City by study group, October 2020-June 2021.**

| | | Total N = 883 | Incident n = 55 | Prevalent n = 235 | Non-cases n = 593 | P value [a] |
|---|---|---|---|---|---|---|
| Sex, n (%) [b] | Female | 620 (70) | 37 (67) | 158 (68) | 425 (72) | 0.44 |
| | Male | 262 (30) | 18 (33) | 76 (32) | 168 (28) | |
| Age (Median, IQR) | | 36 (28–46) | 34 (27–44) | 36 (28–45) | 36 (28–46) | 0.57 |
| Institution, n (%) | INER | 548 (62) | 36 (65) | 115 (49) | 397 (67) | **<0.001** |
| | INCMNSZ | 335 (38) | 19 (35) | 120 (51) | 196 (33) | |
| State of residency, n (%) | Mexico City | 752 (86) | 49 (89) | 187 (81) | 516 (88) | 0.08 |
| | State of Mexico | 100 (11) | 4 (7) | 34 (15) | 62 (10) | |
| | Other | 23 (3) | 2 (4) | 10 (4) | 11 (2) | |
| Municipality, n (%) | Tlalpan | 283 (38) | 17 (35) | 65 (35) | 201 (39) | 0.16 |
| | Coyoacán | 108 (14) | 6 (12) | 22 (12) | 80 (15) | |
| | Xochimilco | 71 (10) | 4 (8) | 26 (14) | 41 (8) | |
| | Iztapalapa | 63 (8) | 7 (14) | 11 (6) | 45 (9) | |
| | Other [c] | 223 (30) | 15 (31) | 61 (33) | 147 (29) | |
| Occupation, n (%) | Physician | 212 (24) | 16 (29) | 47 (20) | 149 (25) | **0.005** |
| | Nurse | 244 (28) | 16 (29) | 76 (33) | 152 (26) | |
| | Lab Technician | 38 (4) | 0 (0) | 16 (7) | 22 (4) | |
| | Administrative | 195 (22) | 19 (35) | 44 (19) | 132 (22) | |
| | Other | 189 (22) | 4 (7) | 50 (21) | 135 (23) | |
| Previous COVID-19 diagnosis, n (%) | Yes | 187 (21) | 5 (9) | 163 (70) | 19 (3) | **<0.001** |
| | No | 694 (79) | 50 (91) | 71 (30) | 573 (97) | |
| Contact with any person with COVID-19, n (%) [d] | Yes | 480 (54) | 35 (64) | 129 (55) | 316 (53) | 0.34 |
| | No/Unknown | 401 (46) | 20 (36) | 105 (45) | 276 (47) | |
| Handling of biological specimens, n (%) | Yes | 326 (37) | 21 (38) | 97 (41) | 208 (35) | 0.23 |
| | No/Unknown | 555 (63) | 34 (62) | 137 (59) | 384 (65) | |
| Contact with COVID-19 patients, n (%) | Frequently | 367 (42) | 21 (38) | 113 (48) | 233 (39) | 0.06 |
| | Never/Ocassionally | 514 (58) | 34 (62) | 121 (52) | 359 (61) | |
| Use of PPE, n (%) | Always/Generally | 812 (92) | 52 (95) | 208 (89) | 552 (93) | 0.09 |
| | Sometimes/Never | 69 (8) | 3 (5) | 26 (11) | 40 (7) | |
| Use of face mask, n (%) | Always/Generally | 875 (99) | 55 (100) | 231 (99) | 589 (99) | 0.39 |
| | Sometimes/Never | 6 (1) | 0 (0) | 3 (1) | 3 (1) | |
| Hand washing, n (%) | Always/Generally | 879 (100) | 55 (100) | 233 (100) | 591 (100) | 0.73 |
| | Sometimes/Never | 2 (0) | 0 (0) | 1 (0) | 1 (0) | |
| Use of public transport, n %) | Yes | 426 (48) | 32 (58) | 121 (52) | 273 (46) | 0.11 |
| | No | 455 (52) | 23 (42) | 113 (48) | 319 (54) | |
| Vaccinated, n (%) [e] | Yes | 597 (68) | 40 (73) | 171 (73) | 386 (65) | 0.07 |
| | No | 286 (32) | 15 (27) | 64 (27) | 207 (35) | |

INCMNSZ, National Institute of Medical Sciences and Nutrition; INER, National Institute of Respiratory Diseases; PPE, Personal Protection Equipment

[a] Chi-square test considering incident, prevalent and non-cases, two-sided P values are shown

[b] Column percentages are shown in all cases

[c] Includes the remaining municipalities in Mexico City and municipalities in other states.

[d] Suspected or confirmed cases during the last 15 days.

[e] Starting from late December 2020, as per national policy, until last follow up visit.

## Antibody prevalence estimation

At the end of the study, 290 participants had a positive result in any of the antibody tests, yielding an overall prevalence in the study period, adjusted by sensitivity and specificity of the

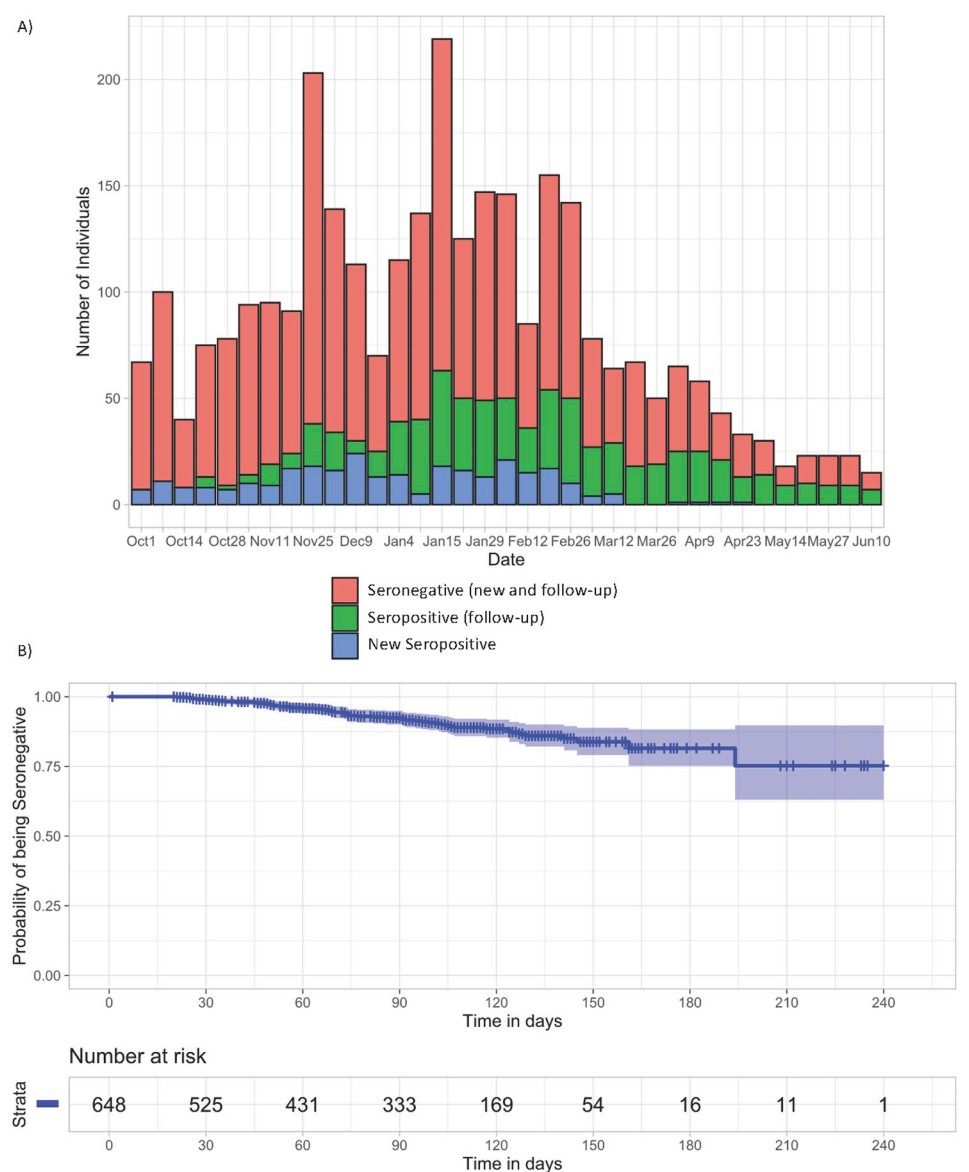

**Fig 1. Variation of enrolment and SARS-CoV-2 seropositivity rate along the study period in healthcare workers of the two largest COVID-19 referral hospitals in Mexico City, October 2020-June 2021.** A. Sampling distribution along the study period (October 2020-June 2021). New seropositive participants correspond to those having a positive serological test result in the first visit; seropositive follow-up corresponds to those having a positive serological test from the second visit onwards; seronegatives include both new and follow-up. B. Kaplan-Meier curve showing the unadjusted probability of being seronegative along time.

antibody test, of 33.5%. There were 235/290 positive tests (81.9%) at baseline (prevalent cases), and the remaining 55/290 (18.9%) seroconverted during the follow up period (incident cases). The median time between visits was 29 days (IQR: 23–40). The total observation time was 76,392 days (2,546.4 months) in 883 participants. There were 774 (88%) participants with at least two follow-up test results, 662 (70%) had at least three, and only a third completed the planned five visits (280/883, 32%) (S1 Table). Fig 1A shows variation of enrolment and seropositivity rate along the study period.

**Table 2. Characteristics of prevalent cases and associated risks of having a positive serological test at baseline in healthcare workers of the two largest COVID-19 referral hospitals in Mexico City, October 2020-June 2021.**

| | | n | (%) [a] | OR | 95% CI | aOR | 95% CI |
|---|---|---|---|---|---|---|---|
| Sex | Male | 76/262 | (29) | | | Reference | |
| | Female | 158/620 | (25) | 0.84 | 0.61–1.16 | 0.86 | 0.58–1.29 |
| Age | N/A | N/A | | 1.00 | 0.98–1.01 | 1.00 | 0.98–1.02 |
| Municipality | Tlalpan | 65/283 | (23) | | | Reference | |
| | Coyoacán | 22/108 | (20) | 0.86 | 0.50–1.48 | 0.93 | 0.52–1.66 |
| | Xochimilco | 26/71 | (37) | **1.94** | **1.11–3.38** | **2.03** | **1.09–3.79** |
| | Iztapalapa | 11/63 | (17) | 0.71 | 0.35–1.44 | 0.69 | 0.32–1.51 |
| | Other [b] | 61/223 | (27) | 1.26 | 0.84–1.89 | 1.31 | 0.85–2.02 |
| Institution | INER | 115/548 | (21) | | | Reference | |
| | INCMNSZ | 120/335 | (36) | **2.10** | **1.55–2.85** | **2.24** | **1.54–3.25** |
| Occupation | Physician | 47/212 | (22) | | | Reference | |
| | Nurse | 76/244 | (31) | 1.59 | 1.04–2.42 | 1.02 | 0.57–1.81 |
| | Lab Technician | 16/38 | (42) | **2.55** | **1.24–5.25** | **4.38** | **1.75–10.93** |
| | Administrative | 44/195 | (23) | 1.02 | 0.64–1.63 | 1.18 | 0.64–2.17 |
| | Other | 50/189 | (26) | 1.26 | 0.80–2.00 | 1.40 | 0.79–2.50 |
| Contact with any person with COVID-19 | | 129/480 | (27) | 1.04 | 0.77–1.40 | 0.78 | 0.52–1.19 |
| Handling of biological specimens | | 97/326 | (30) | 1.29 | 0.95–1.75 | 1.09 | 0.67–1.77 |
| Contact with COVID-19 patients | | 113/367 | (31) | **1.44** | **1.07–1.95** | 1.61 | 0.99–2.61 |
| Use of PPE | | 208/812 | (26) | **0.57** | **0.34–0.95** | 0.60 | 0.32–1.13 |
| Use of face mask | | 231/875 | (26) | 0.36 | 0.07–1.79 | 0.44 | 0.05–3.91 |
| Use of public transport | | 121/426 | (28) | 1.20 | 0.89–1.62 | 1.27 | 0.87–1.85 |

aOR, adjusted odds ratio; OR, crude odds ratio; CI, confidence interval; INCMNSZ, National Institute of Medical Sciences and Nutrition; INER, National Institute of Respiratory Diseases; N/A, not applicable; PPE, Personal Protection Equipment.

[a] Row percentages are shown.

[b] Includes the remaining municipalities in Mexico City and municipalities in other states. Only the variables shown in the table were included in the multivariable model.

## Risk factors in prevalent cases

Among the 235 cases with positive antibody test at their first visit, 163 (70%) reported having previously been diagnosed with SARS-CoV-2 infection (either by RT-PCR or rapid antigen test) (Table 1), with a median time between diagnosis and sample donation of 103 days (IQR: 45–314). As expected, having previous diagnosis of SARS-CoV-2 infection was strongly associated with having a positive antibody test (odds ratio [OR] = 63.3, 95% confidence interval [CI]: 38.5, 104.1); thus, we excluded this variable from the multivariable model due to collinearity (Table 2). The observed seroprevalence at baseline in physicians was 22%, 31% in nurses, 42% in laboratory technicians, and 23% in administrative staff (Table 2). When considering prevalent cases exclusively, physicians and nurses had a higher frequency of contact with COVID-19 patients, and handled biological specimens more often than administrative staff ($P<0.001$ in both cases) (S3 Table). The most frequent symptoms in prevalent cases were fatigue (65%), headache (65%), and myalgia (63%) (S2 Table). Nearly all symptoms assessed were more frequent in prevalent cases than in non-cases ($P<0.01$ in all cases) (S2 Table).

Persons working at INCMNSZ vs. INER (adjusted odds ratio [aOR] = 2.2, 95% CI: 1.5, 3.3), living in the municipality of Xochimilco vs. Tlalpan in Mexico City (aOR: 2.0, 95% CI: 1.1, 3.8), and working as laboratory technician in comparison to physicians (aOR: 4.4, 95% CI: 1.8, 10.9) had increased odds of having a positive test in the first visit. Associations were adjusted

by sex, age, recent contact with persons with COVID-19, handling biological specimens of patients with COVID-19, contact with COVID-19 patients, appropriate use of PPE, and using public transportation for commuting to work (Table 2).

### Risk factors in incident cases

After excluding the 235 participants with positive tests in the first visit (prevalent cases), the remaining 55 with positive tests seroconverted during the 55,008 days (1,833.6 months) of follow-up, resulting in an overall incidence of 3.0 cases per 100 persons-month. Of the 55 incident cases, 30 were identified at the second visit, 15 at the third, seven at the fourth, and three at the fifth (Fig 1B; S1 Table). Using a Cox regression model stratified by institution, contact with any suspected or confirmed COVID-19 case in the previous 15 days increased the hazard of becoming an incident case (HR = 2.5, 95% CI: 1.2, 5.0), while having a different occupation to physician, nurse or administrative staff, decreased this hazard (HR = 0.23, 95% CI: 0.06, 0.83) (Table 3). No additional hazard associations were observed including use of PPE, contact with COVID-19 patients, or use of public transport.

The most frequently observed symptoms among incident cases were fatigue (60%) and headache (51%) (S2 Table). Fever, myalgia, diarrhea, anosmia, and ageusia were significantly less frequent in incident cases, compared to prevalent cases ($P<0.05$ in all cases) (S2 Table). Contact with COVID-19 patients was significantly less frequent in incident cases among administrative staff, compared to physicians and nurses ($p<0.05$), while contact with any person with COVID-19, both at work and in the community, was frequent in all occupations: physicians (50%), nurses (88%), administrative staff (42%) (S4 Table).

### Discussion

To our knowledge, this is the first report on seroprevalence of SARS-CoV-2 infection in healthcare workers in a Mexican setting. The participating institutions are the largest tertiary

**Table 3. Cox model on risks associated with being an incident case in healthcare workers of the two largest COVID-19 referral hospitals in Mexico City, October 2020-June 2021.**

| Variable | Categories | n [a] | HR | 95% CI |
|---|---|---|---|---|
| Sex | Female | 37 | | Reference |
| | Male | 18 | 0.65 | 0.34–1.24 |
| Age | | | 0.80 | 0.58–1.10 |
| Occupation | Physician | 16 | | Reference |
| | Nurse | 16 | 0.94 | 0.40–2.19 |
| | Administrative | 19 | 1.55 | 0.68–3.52 |
| | Other | 4 | **0.23** | **0.06–0.83** |
| Contact with any person with COVID-19 [b] | No/Unknown | 7 | | Reference |
| | Yes | 48 | **2.45** | **1.21–5.00** |
| Contact with patients with COVID-19 | Never/Occasionally | 17 | | Reference |
| | Frequently | 38 | 1.18 | 0.55–2.50 |
| Use of PPE | No | 2 | | |
| | Yes | 53 | 0.38 | 0.14–1.01 |
| Use of public transport | No | 30 | | Reference |
| | Yes | 25 | 1.30 | 0.68–2.50 |

HR, Hazard ratio; CI, confidence interval; PPE, Personal Protection Equipment

[a] Total incident cases = 55

[b] Suspected or confirmed during the last 15 days. Only the variables shown in the table were included in the multivariable model.

care, referral centers in Mexico City, transformed into COVID-19-only facilities to care for severely ill patients. Our results show a notably high seroprevalence to SARS-CoV-2 in healthcare workers at these institutions with nearly a third of the participants showing exposure to SARS-CoV-2 at any time during the study period, most of them at their first visit. Prevalent cases (those positive at the first visit) appeared to be associated with both occupational and community exposure to SARS-CoV-2 infection, including institution, occupation, and place of residence; while incident cases were strongly associated with community exposure, mainly contact with persons with COVID-19 outside of the hospital.

The institutions selected for the study were amongst the ones with better access to resources to respond to the sanitary crisis, which could decrease occupational hazard in the study population and makes it difficult to extrapolate the results to other institutions in the country and even in Mexico City, with generally poorer infrastructure and access to medical supplies, and training. Indeed, in our study, access to PPE and personnel training were good in general, as suggested by most participants reporting high frequency of PPE use and compliance with contingency sanitary measures. Despite this, the observed overall seroprevalence (33.5%) is amongst the highest reported in the literature [32, 36, 39]. This observation could be explained by several factors. First, the study period (October 2020-June 2021) comprised the second epidemiological wave observed in Mexico during the winter season, with the highest peak of daily cases, both nationwide and in Mexico City [1]. This study was conducted later during the pandemic than other studies with similar and lower prevalence [8–10, 33]; thus, the observed high prevalence may be explained by higher accumulation of cases. Second, our results are consistent with those of two large, national, seroepidemiological surveys, that observed that approximately one third of the general population had been exposed to SARS-CoV-2 by December 2020 [3, 6, 7]. This could be especially true in large urban centers such as Mexico City, which has contributed with a large proportion of the total national cases [1, 3]. It is also consistent with the predominantly community-associated risk observed in this study, especially in incident cases.

We observed that in prevalent cases, place of residence was associated with exposure, with Xochimilco municipality associated with higher seropositivity at the first visit. Regarding occupational hazards for prevalent cases, working at INCMNSZ and being a laboratory technician were associated with higher odds of exposure. The fact that this association was only observed with prevalent cases but not incident cases, could imply that mitigation measures in general and training for healthcare personnel, improved over time in both institutions, but especially at INCMNSZ. The higher risk observed in laboratory technicians cannot be explained by work-associated activities such as frequency of biological specimen handling and contact with COVID-19 patients, which were similar to those reported by physicians and nurses (S3 Table), and could reflect selection bias in this occupational group. Indeed, laboratory technicians showed a higher frequency of previous COVID-19 diagnosis (36.8%), compared to physicians (18.0%), nurses (23.5%) or administrative staff (19%), suggesting a higher enrollment of participants interested in the results of the antibody tests for this occupational group. A recent study reported low nosocomial acquisition of SARS-CoV-2 infection at INER, one of the two participating institutions here, with the implementation of a successful occupational medicine program [42]. This program included intensive training on adequate use of PPE, and follow-up of workers with readily available molecular and antigen testing for SARS-CoV-2. Although we did observe occupational risks of exposure, our results are consistent with this report with an observed predominance of community-associated risk as the study progressed.

We cannot rule out that the high prevalence observed in our study could also reflect selection bias, as participants could have been led to participation by interest in the result of the serological test, having had a high suspicion of previous disease or exposure. Moreover,

participation in the study was much lower than expected, with only 45% (548/1,211) of INER and 14% (335/2,428) of the registered personnel of INCMNSZ enrolled. This could be attributable to lack of interest in participating, little time due to high-work burden, competing research protocols, problems accessing the electronic portal because of poor computational skills or lack of information on the current work (even after multiple communication efforts to promote the study). Indeed, we observed some differences between persons who did not attend (n = 246) versus those who attended (n = 883) their appointment for blood collection after registering in the study portal, including lower age, higher percentage of workers from INCMNSZ, higher proportion of nurses, higher proportion of persons that reported handling biological specimens, having frequent contact with patients with COVID-19, and using public transport (P<0.05 in all cases; S5 Table).

Our study has several limitations worth mentioning. First, the enrolment method was susceptible to selection bias, as mentioned above. Second, although the study was performed in the two largest COVID-19 facilities in Mexico City, the fact that only two centers were included could affect representativeness of the study. Also, the participating institutions were among the ones with better access to medical supplies, PPE and training, and the results may not be representative of other COVID-19 centers in Mexico or even in Mexico City. Third, regarding possible risk factors and symptoms associated with prevalent cases, information bias could exist, as participants were asked to recall information of events that could have happened since March 2020, during the enrolment period starting in October 2020. For incident cases, this memory bias is expected to be reduced, with the questionnaire referring to events happening in the previous 15 days to sample collection. Fourth, the high rate of follow-up loses, with only a third of the participants completing the total of follow up visits planed could affect incidence estimations. Finally, as discussed in a recently published study [43], using anti-N antibody tests for population-based seroprevalence studies may result in the underestimation of prevalence due to waning of these antibodies over time in comparison with anti-S antibodies. This is a limitation of our study, although the impact of this observation in our setting would need to be assessed given that the first cases of SARS-CoV-2 infection in Mexico were reported in March 2020 and enrollment in our study began in October 2020, coinciding with the onset of the second wave that significantly increased infection incidence [1]. Thus, we would expect a high proportion of the detected infections at baseline to be relatively recent. On the other hand, measuring anti-N antibodies allows to distinguish previously infected individuals from vaccinated individuals (for most approved vaccines). This is important given that vaccination efforts for healthcare personnel in Mexico started by the end of 2020. Nevertheless, even considering a possible underestimation, the prevalence rate was notably high in the examined population.

## Conclusion

We observed a high rate of exposure to SARS-CoV-2 in healthcare workers in Mexico City, with overall seroprevalence in the study period of 33.5% from October 2020 to June 2021. The incidence rate was 3.0 cases/100 person-months. Our study suggests that the highest risk of exposure in healthcare workers in this setting occurred in the community, although occupational risk was also observed, especially in prevalent cases. Prevalent cases were associated with institution, occupation, and place of residence, while incident cases with recent contact with persons with COVID-19. No associations of a positive antibody test and contact with COVID-19 patients or handling of biological specimens were observed. Moreover, a high proportion of the participants reported good compliance with contingency measures and good access to PPE. Our observations are consistent with successful occupational medicine programs for

SARS-CoV-2 infection control in the participating institutions but suggest the need to strengthen mitigation strategies in the community.

## Supporting information

**S1 Appendix. Computer-assisted self-administered interview questions.**
(DOCX)

**S2 Appendix. De-identified database.**
(DTA)

**S1 Table. Variation in the number of participants at baseline and follow-up along the study period.**
(DOCX)

**S2 Table. Symptoms reported by exposure group in healthcare workers of the two largest COVID-19 referral hospitals in Mexico City, October 2020-June 2021.**
(DOCX)

**S3 Table. Risk factors by occupation in prevalent cases, October 2020-June 2021.**
(DOCX)

**S4 Table. Risk factors by occupation in incident cases, October 2020-June 2021.**
(DOCX)

**S5 Table. Differences between persons who did and did not attend at least one visit for blood sample donation for antibody testing that registered in the study database of healthcare workers of the two largest COVID-19 referral hospitals in Mexico City, October 2020-June 2021.**
(DOCX)

## Acknowledgments

We thank all participants for their time and generosity. We acknowledge Edna Rodríguez-Aguirre, Martha Huertas, and the sample collection staff at INER and INCMNSZ for their contribution to this work.

## Author Contributions

**Conceptualization:** Pablo F. Belaunzarán-Zamudio, Gustavo Reyes-Terán, Juan G. Sierra-Madero, Santiago Ávila-Ríos.

**Data curation:** Vanessa Dávila-Conn, Yanink N. Caro-Vega, Héctor E. Paz-Juárez.

**Formal analysis:** Vanessa Dávila-Conn, Maribel Soto-Nava, Yanink N. Caro-Vega.

**Funding acquisition:** Gustavo Reyes-Terán, Santiago Ávila-Ríos.

**Investigation:** Vanessa Dávila-Conn, Maribel Soto-Nava, Pedro García-Esparza, Daniela Tapia-Trejo, Marissa Pérez-García.

**Methodology:** Maribel Soto-Nava, Yanink N. Caro-Vega, Daniela Tapia-Trejo, Marissa Pérez-García.

**Project administration:** Juan G. Sierra-Madero, Arturo Galindo-Fraga, Santiago Ávila-Ríos.

**Software:** Héctor E. Paz-Juárez.

**Supervision:** Vanessa Dávila-Conn, Maribel Soto-Nava, Santiago Ávila-Ríos.

**Validation:** Santiago Ávila-Ríos.

**Writing – original draft:** Santiago Ávila-Ríos.

**Writing – review & editing:** Vanessa Dávila-Conn, Yanink N. Caro-Vega, Pablo F. Belaunzarán-Zamudio, Gustavo Reyes-Terán, Juan G. Sierra-Madero, Arturo Galindo-Fraga.

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
