## [Decision Letter · Decision Letter 0]

10 Dec 2021

PONE-D-21-27465Seroepidemiology of SARS-CoV-2 in healthcare personnel working at the largest tertiary COVID-19 referral hospitals in Mexico City.PLOS ONE

Dear Dr. Avila-Rios,

Thank you for submitting your manuscript to PLOS ONE. After careful consideration, we feel that it has merit but does not fully meet PLOS ONE’s publication criteria as it currently stands. Therefore, we invite you to submit a revised version of the manuscript that addresses the points raised during the review process.

We look forward to receiving your revised manuscript.

Kind regards,

Leeberk Raja Inbaraj, MD

Academic Editor

PLOS ONE

Journal Requirements:

“This work was supported by Consejo Nacional de Ciencia y Tecnología (CONACyT) (Fondo FORDECYT-PRONACES) and the Mexican Government (Programa Presupuestal P016; Anexo 13 del Decreto del Presupuesto de Egresos de la Federación) to SAR. Funders had no role in study design, data collection, analysis or interpretation, writing of the report and decision to submit for publication.”

“This work was supported by Consejo Nacional de Ciencia y Tecnología (CONACyT) (Fondo FORDECYT-PRONACES) and the Mexican Government (Programa Presupuestal P016; Anexo 13 del Decreto del Presupuesto de Egresos de la Federación) to SAR. Funders had no role in study design, data collection, analysis or interpretation, writing of the report and decision to submit for publication.”

Reviewers' comments:

Reviewer's Responses to Questions

**Comments to the Author**

1. Is the manuscript technically sound, and do the data support the conclusions?

Reviewer #1: Yes

Reviewer #2: Yes

2. Has the statistical analysis been performed appropriately and rigorously? 

Reviewer #1: Yes

Reviewer #2: Yes

3. Have the authors made all data underlying the findings in their manuscript fully available?

Reviewer #1: Yes

Reviewer #2: Yes

4. Is the manuscript presented in an intelligible fashion and written in standard English?

Reviewer #1: Yes

Reviewer #2: Yes

5. Review Comments to the Author

Reviewer #1: This is an important study done during difficult times, among healthcare personnel and the work done is appreciated.

The study brings out the main point that the exposure in this healthcare setting pointed toward community rather than occupational risk. Some comments are:

1. Since the study reports a prevalence value among healthcare staff, it may be important to mention that since the Nucleocapsid antigen was used, there could be underestimation of the prevalence as compared to the assays using the Spike protein as per some reports (eg: Changes in SARS-CoV-2 Spike versus Nucleoprotein Antibody Responses Impact the Estimates of Infections in Population-Based Seroprevalence Studies | Journal of Virology (asm.org)).

2. Although lab technicians are stated to be associated with higher risk at the first visit, and this is mentioned as a significant result in the abstract, no details are available about the specific type of work done related to exposure (sample collection or testing etc), use and availability of PPE etc for this group and on the possible reasons for the same especially when compared to doctors and nurses. Or was it a selection bias alone?

3. Line 181: In the models, participants

“182 were censored at the time of their last test, when they did not complete the originally planned 183 five tests and had negative results in all their samples.”

This statement is not clear. Were they removed completely?

4. Line 321 – The use of the word “posterior “is confusing. Does it mean “behind/before” or after?

Reviewer #2: Abstract

• Details regarding the study participants can be added

• consent for participation and regular blood testing can be added

• Study duration and frequency of the blood tests done can be added

• What is the target sample size?

Introduction

• Information on the reported burden (how many cases were reported during the first wave) and rationale of seroprevalence can be linked

• More content needs to be added on the need and process of conducting seroprevalence studies.

• Need a clear description on what is the knowledge/ research gap and how current study tries to address it.

Methods

• The rationale for conducting this study is not clearly mentioned anywhere, especially What is the rationale of including all types of health workers, did the authors assume that the risk was same for all the workers. If it is different risk, how did they analyze the sample size for each risk level?

• Who had access to CASI data? How was the data security maintained?

• Line no. 146: please add reference number of approval from ethical committee

Results:

• Line number 233 to 235

• It is not clear as to whether it is a Figure title or started with explanation?

• What do you mean by prevalent cases as written in line number 250? The description of prevalent cases is given only in line number 275. It would be better to add it in line number 250 for better clarity

• Line No (255-262): Too lengthy sentence and not clear

• Table 2: What are the factors adjusted for Table 2? Please mention adjusted variables in the footnote of the table

Discussion

• Line No 343.:What are the reasons for lower participation rate in both the study centers?

• How the data privacy of the health care staff was maintained across the study centers? This might be one of the reasons for non-participation too. How does the study overcome this issue? Please describe.

• Line number 233 to 235: It is not clear as to whether it is a Figure title or started with explanation?

6. PLOS authors have the option to publish the peer review history of their article (what does this mean?). If published, this will include your full peer review and any attached files.

Reviewer #1: No

Reviewer #2: No

---

## [Author Response · Author response to Decision Letter 0]

17 Jan 2022

Journal Requirements:

A: The manuscript has been formatted to meet PLoS One’s style requirements.

A: The questionnaire has been added as a supplementary file. 

“This work was supported by Consejo Nacional de Ciencia y Tecnología (CONACyT) (Fondo FORDECYT-PRONACES) and the Mexican Government (Programa Presupuestal P016; Anexo 13 del Decreto del Presupuesto de Egresos de la Federación) to SAR. Funders had no role in study design, data collection, analysis or interpretation, writing of the report and decision to submit for publication.”

“This work was supported by Consejo Nacional de Ciencia y Tecnología (CONACyT) (Fondo FORDECYT-PRONACES) and the Mexican Government (Programa Presupuestal P016; Anexo 13 del Decreto del Presupuesto de Egresos de la Federación) to SAR. Funders had no role in study design, data collection, analysis or interpretation, writing of the report and decision to submit for publication.”

A: We confirm that the information included in the Funding Statement is correct. We have removed this information from the Acknowledgements section.

b) If there are no restrictions, please upload the minimal anonymized data set necessary to replicate your study findings as either Supporting Information files or to a stable, public repository and provide us with the relevant URLs, DOIs, or accession numbers. For a list of acceptable repositories, please seehttp://journals.plos.org/plosone/s/data-availability#loc-recommended-repositories.

A: We have included the anonymized dataset used for the analyses as supporting information.

A: The full ethics statement is now included in the Methods, including data on the IRB who approved the study.

A: The ethics statement has been moved to the Methods and removed from other sections of the manuscript. 

  Reviewers' comments:  Reviewer's Responses to Questions

 1. Is the manuscript technically sound, and do the data support the conclusions?  The manuscript must describe a technically sound piece of scientific research with data that supports the conclusions. Experiments must have been conducted rigorously, with appropriate controls, replication, and sample sizes. The conclusions must be drawn appropriately based on the data presented. 

Reviewer #1: Yes

Reviewer #2: Yes

2. Has the statistical analysis been performed appropriately and rigorously? 

Reviewer #1: Yes

Reviewer #2: Yes

3. Have the authors made all data underlying the findings in their manuscript fully available?  The PLOS Data policy requires authors to make all data underlying the findings described in their manuscript fully available without restriction, with rare exception (please refer to the Data Availability Statement in the manuscript PDF file). The data should be provided as part of the manuscript or its supporting information, or deposited to a public repository. For example, in addition to summary statistics, the data points behind means, medians and variance measures should be available. If there are restrictions on publicly sharing data—e.g. participant privacy or use of data from a third party—those must be specified.

Reviewer #1: Yes

Reviewer #2: Yes

4. Is the manuscript presented in an intelligible fashion and written in standard English?  PLOS ONE does not copyedit accepted manuscripts, so the language in submitted articles must be clear, correct, and unambiguous. Any typographical or grammatical errors should be corrected at revision, so please note any specific errors here.

Reviewer #1: Yes

Reviewer #2: Yes

5. Review Comments to the Author  Please use the space provided to explain your answers to the questions above. You may also include additional comments for the author, including concerns about dual publication, research ethics, or publication ethics. (Please upload your review as an attachment if it exceeds 20,000 characters)

Comments to the Author

Reviewer #1: 

This is an important study done during difficult times, among healthcare personnel and the work done is appreciated. 

The study brings out the main point that the exposure in this healthcare setting pointed toward community rather than occupational risk. Some comments are:  1. Since the study reports a prevalence value among healthcare staff, it may be important to mention that since the Nucleocapsid antigen was used, there could be underestimation of the prevalence as compared to the assays using the Spike protein as per some reports (eg: Changes in SARS-CoV-2 Spike versus Nucleoprotein Antibody Responses Impact the Estimates of Infections in Population-Based Seroprevalence Studies | Journal of Virology (asm.org)).

A: We thank the Reviewer for this observation. We have included an additional paragraph in the Discussion to mention this point as a limitation of the study: “Finally, as discussed in a recently published study [43], using anti-N antibody tests for population-based seroprevalence studies may result in the underestimation of prevalence due to waning of these antibodies over time in comparison with anti-S antibodies. This is a limitation of our study, although the impact of this observation in our setting would need to be assessed given that the first cases of SARS-CoV-2 infection in Mexico were reported in March 2020 and enrollment in our study began in October 2020, coinciding with the onset of the second wave that significantly increased infection incidence [1, 5]. Thus, we would expect a high proportion of the detected infections at baseline to be relatively recent. On the other hand, measuring anti-N antibodies allows to distinguish previously infected individuals from vaccinated individuals (for most approved vaccines). This is important given that vaccination efforts for healthcare personnel in Mexico started by the end of 2020. Nevertheless, even considering a possible underestimation, the prevalence rate was notably high in the examined population.” 

 2. Although lab technicians are stated to be associated with higher risk at the first visit, and this is mentioned as a significant result in the abstract, no details are available about the specific type of work done related to exposure (sample collection or testing etc), use and availability of PPE etc for this group and on the possible reasons for the same especially when compared to doctors and nurses. Or was it a selection bias alone?

A: The specific type of work of laboratory technicians was not collected as part of the metadata. However, we do have information on access to PPE, contact with COVID-19 patients and handling of biological specimens from COVID-19 patients (S3 Table). The self-reported frequency of biological specimen handling and contact with COVID-19 patients among laboratory technicians was not significantly different to that reported by nurses and physicians and thus, we would expect a higher significance of community-associated risk. However, this observation could also reflect selection bias in this group. Indeed, the frequency of previous COVI-19 diagnosis in laboratory technicians was higher than in physicians, nurses and administrative staff, suggesting a higher enrollment of persons interested in the results of the antibody tests for participants of this occupational group. This has been added to the Discussion.

3. Line 181: In the models, participants “182 were censored at the time of their last test, when they did not complete the originally planned 183 five tests and had negative results in all their samples.” This statement is not clear. Were they removed completely?

A: All the participants were included in the prevalence analysis. Participants with a positive result in their first sample were excluded from the incidence analysis and were considered prevalent cases. Participants who did not complete the five follow-up visits contributed person-time from all the available tests with negative results, until the first positive result or until their last available negative result. We have included the following sentence in the text to improve clarity: “In the models, participants with negative results in all their follow-up samples were censored at the time of their last available test.”

 4. Line 321 – The use of the word “posterior “is confusing. Does it mean “behind/before” or after?

A: We have reworded the sentence in order to make it clearer: “This study comprises a later time period to that reported in other studies with similar and lower prevalence [8-10, 33].”

Reviewer #2: 

Abstract • Details regarding the study participants can be added • consent for participation and regular blood testing can be added • Study duration and frequency of the blood tests done can be added • What is the target sample size?

A: Considering word limit, the abstract has been modified to include the details requested by the Reviewer. Study duration and frequency of blood tests were already included in the original version.  Introduction • Information on the reported burden (how many cases were reported during the first wave) and rationale of seroprevalence can be linked • More content needs to be added on the need and process of conducting seroprevalence studies. • Need a clear description on what is the knowledge/ research gap and how current study tries to address it. 

A: We believe that the original version of the manuscript already includes several of the points raised by the Reviewer to explain the rationale of the study as well as the knowledge gap being addressed: 

1. Higher risk of exposure among health workers than the general population: Lines 77-79

2. Association of exposure with activities at work and working conditions: Lines 81-84

3. Wide variation in reported seroprevalence in healthcare workers across studies worldwide: Lines 84-86.

4. Unknown seroprevalence for SARS-CoV-2 in healthcare workers in the Mexican setting: Lines 102-103.

5. Advantages of conducting seroprevalence studies in healthcare workers to better understand occupational hazards and improve prevention measures: Lines 97-102

In order to improve this justification, we have added some additional content to the Introduction, as suggested, including an update of the epidemiological background (lines 67-71) and rationale of the study (lines 104-106; lines 110-112).

 Methods • The rationale for conducting this study is not clearly mentioned anywhere, especially What is the rationale of including all types of health workers, did the authors assume that the risk was same for all the workers. If it is different risk, how did they analyze the sample size for each risk level?

A: We believe that the original version of the manuscript includes several points to explain the rationale of the study, as detailed above. 

In order to improve this justification, and following the Reviewer’s suggestion, we have added additional text in the Methods, explaining the rationale for including all types of healthcare workers: “We invited all healthcare workers of the participating institutions to enroll, regardless of occupation (staff physicians, physicians in training, nurses, laboratory technicians, researchers, students, housekeeping, maintenance, security and administrative staff). We included all occupations, given the hypothesized overall increased risk of exposure in the working environment due to the concentration of severely ill persons in the participating institutions, as well as the possibility of frequent contact of co-workers with different risk of exposure to SARS-CoV-2 according to their activities. Participants were enrolled between October 2020 and June 2021 and followed-up for up to five visits, separated at least 21 days one from each other. We collected blood samples for antibody testing and applied structured questionnaires for sociodemographic and occupational and community risk exposure information at each visit. We evaluated the risk of exposure to SARS-CoV-2 using multivariable models including the type of occupation and potential community and work risk variables.” 

Sample size for different risk levels according to occupation were not calculated. Instead, we analyzed the independent and potential association of each work and community risk variable (including occupation) with SARS-CoV-2 exposure using multivariable logistic models. Stratifying by occupation was not possible, given the low number of participants in some categories. 

 • Who had access to CASI data? How was the data security maintained?

A: We have added the following information to the “Data collection” section of the Methods: “Data was stored in a local secured server. Only key research personnel and IT staff had access to the data and databases used for the analyses were previously de-identified. Each participant had access to a portal containing their individually scheduled appointments and the results from serological tests. Data privacy and access policies were explained to and accepted by the participants during the informed consent process and before answering the CASI”. 

 • Line no. 146: please add reference number of approval from ethical committee 

A: An Ethics Statement section has been added, including registry numbers of the Institutional Review Boards, as well as reference numbers of the approved protocol. 

Results: • Line number 233 to 235 • It is not clear as to whether it is a Figure title or started with explanation?

A: We apologize for the confusion. The Journal requires Figure Legends to be inserted across the manuscript. This paragraph includes the title and legend for Figure 1. We have corrected formatting issues in order to make it more understandable. 

 • What do you mean by prevalent cases as written in line number 250? The description of prevalent cases is given only in line number 275. It would be better to add it in line number 250 for better clarity

A: We define prevalent, incident and non-cases in the Methods (Statistical analyses section). For better clarity, we have added the definition in the suggested line: “Considering only prevalent cases (persons with a positive antibody test result in their first visit; see Methods)” 

  • Line No (255-262): Too lengthy sentence and not clear • Table 2: What are the factors adjusted for Table 2? Please mention adjusted variables in the footnote of the table

A: We have reworded the paragraph, separating sentences to improve clarity. We have added a footnote in Tables 2 and 3, explaining that only variables shown in the table were included in the multivariable models. This information is also included in the Statistical Analysis section of the Methods.

 Discussion • Line No 343.: What are the reasons for lower participation rate in both the study centers?

A: The reasons for low participation rate and its impact as a possible bias in the study are discussed in the original manuscript (lines 362-373 of the revised version). This section of the text was kept as originally written in the revised version: “Participation in the study was much lower than expected, with only 45% (548/1,211) and 14% (335/2,428) of the registered personnel of INER and INCMNSZ enrolled. This could be attributable to lack of interest in participating, little time due to high work burden, competing research protocols, problems accessing the electronic portal because of poor computational skills or lack of information on the current work (even after multiple communication efforts to promote the study). Indeed, we observed some differences between persons who did not attend (n=246) versus those who attended (n=883) their appointment for blood collection after registering in the study portal, including lower age, higher percentage of workers from INCMNSZ, higher proportion of nurses, higher proportion of persons that reported handling biological specimens, having frequent contact with patients with COVID-19, and using public transport (P<0.05 in all cases; S5 Table).“

 • How the data privacy of the health care staff was maintained across the study centers? This might be one of the reasons for non-participation too. How does the study overcome this issue? Please describe. 

A: We do not believe that data privacy could be a reason for non-participation. The databases compiling information about the participating healthcare personnel were only available for key research personnel and IT staff and databases for the analyses were previously de-identified. Data privacy and access policy were stated in the informed consent and before providing access to the CASI in the portal. All participants agreed with data privacy policies before providing answers. These points have been added to the “Data collection” section of the Methods.

• Line number 233 to 235: It is not clear as to whether it is a Figure title or started with explanation?

A: We apologize for the confusion. As explained above, the Journal requires Figure Legends to be inserted across the manuscript. This paragraph includes the title and legend for Figure 1. We have corrected formatting issues in order to make it more understandable. 

6. PLOS authors have the option to publish the peer review history of their article (what does this mean?). If published, this will include your full peer review and any attached files.  Do you want your identity to be public for this peer review? For information about this choice, including consent withdrawal, please see our Privacy Policy.

Reviewer #1: No

Reviewer #2: No

---

## [Decision Letter · Decision Letter 1]

7 Feb 2022

PONE-D-21-27465R1Seroepidemiology of SARS-CoV-2 in healthcare personnel working at the largest tertiary COVID-19 referral hospitals in Mexico City.PLOS ONE

Dear Dr. Avila-Rios,

Thank you for submitting your manuscript to PLOS ONE. After careful consideration, we feel that it has merit but does not fully meet PLOS ONE’s publication criteria as it currently stands. Therefore, we invite you to submit a revised version of the manuscript that addresses the points raised during the review process.

We look forward to receiving your revised manuscript.

Kind regards,

Leeberk Raja Inbaraj, MD

Academic Editor

PLOS ONE

Journal Requirements:

Reviewers' comments:

Reviewer's Responses to Questions

**Comments to the Author**

1. If the authors have adequately addressed your comments raised in a previous round of review and you feel that this manuscript is now acceptable for publication, you may indicate that here to bypass the “Comments to the Author” section, enter your conflict of interest statement in the “Confidential to Editor” section, and submit your "Accept" recommendation.

Reviewer #1: (No Response)

Reviewer #2: All comments have been addressed

2. Is the manuscript technically sound, and do the data support the conclusions?

Reviewer #1: Yes

Reviewer #2: Yes

3. Has the statistical analysis been performed appropriately and rigorously? 

Reviewer #1: Yes

Reviewer #2: Yes

4. Have the authors made all data underlying the findings in their manuscript fully available?

Reviewer #1: Yes

Reviewer #2: Yes

5. Is the manuscript presented in an intelligible fashion and written in standard English?

Reviewer #1: Yes

Reviewer #2: Yes

6. Review Comments to the Author

Reviewer #1: All comments except 3 are fully addressed.

Regarding the earlier comment 3.

Line 181: In the models, participants“182 were censored at the time of their last test, when they did not complete the originally planned 183 five tests and had negative results in all their samples.” This statement is not clear. Were they removed completely?

A: All the participants were included in the prevalence analysis. Participants with a positive result in their first sample were excluded from the incidence analysis and were considered prevalent cases. Participants who did not complete the five follow-up visits contributed person-time from all the available tests with negative results, until the first positive result or until their last available negative result. We have included the following sentence in the text to improve clarity: “In the models, participants with negative results in all their follow-up samples were censored at the time of their last available test.”

From the response to the earlier comment above, it appears that the person time included the time till their last available negative visit if they did not complete five visits.

"Participants who did not complete the five follow-up visits contributed person-time from all the available tests with negative results, until the first positive result or until their last available negative result."

“In the models, participants with negative results in all their follow-up samples were censored at the time of their last available test.”

But does “censored” mean that they were removed? It is confusing if they were removed, whether only the last test was removed and also why they were removed. Or were they removed for calculation of the incident cases ?

Additional comment 1:

At the end of the study, 290 participants had a positive result in any of the antibody tests, yielding 241 an overall prevalence in the study period, adjusted by sensitivity and specificity of the antibody

Line 242 test, of 33.5%. 235 positive cases (81.9%) were identified at baseline (prevalent cases), while the 243 remaining 55 (6.2%) seroconverted during the follow up period (incident cases).

Among the positive cases (290) – the prevalent cases were 235/290 = 81.9%, while the incident cases would be 55/290 which is 18.9 % and not 6.2% as mentioned – is that correct ?

Additional comment 2: Line 68 - spelling error "waive" caused by the omicron variant.

Reviewer #2: Thanks for addressing the earlier comments.

I encourage the authors to please check the manuscript for grammar and sentence structuring before final proof is sent.

7. PLOS authors have the option to publish the peer review history of their article (what does this mean?). If published, this will include your full peer review and any attached files.

Reviewer #1: No

Reviewer #2: **Yes: **Giridhara R Babu

---

## [Author Response · Author response to Decision Letter 1]

11 Feb 2022

Journal Requirements:  Please review your reference list to ensure that it is complete and correct. If you have cited papers that have been retracted, please include the rationale for doing so in the manuscript text, or remove these references and replace them with relevant current references. Any changes to the reference list should be mentioned in the rebuttal letter that accompanies your revised manuscript. If you need to cite a retracted article, indicate the article’s retracted status in the References list and also include a citation and full reference for the retraction notice.   

A: References have been reviewed.

 

Reviewers' comments:  Reviewer's Responses to Questions

Comments to the Author  1. If the authors have adequately addressed your comments raised in a previous round of review and you feel that this manuscript is now acceptable for publication, you may indicate that here to bypass the “Comments to the Author” section, enter your conflict of interest statement in the “Confidential to Editor” section, and submit your "Accept" recommendation.

Reviewer #1: (No Response)

Reviewer #2: All comments have been addressed

2. Is the manuscript technically sound, and do the data support the conclusions?  The manuscript must describe a technically sound piece of scientific research with data that supports the conclusions. Experiments must have been conducted rigorously, with appropriate controls, replication, and sample sizes. The conclusions must be drawn appropriately based on the data presented. 

Reviewer #1: Yes

Reviewer #2: Yes

3. Has the statistical analysis been performed appropriately and rigorously? 

Reviewer #1: Yes

Reviewer #2: Yes

4. Have the authors made all data underlying the findings in their manuscript fully available?  The PLOS Data policy requires authors to make all data underlying the findings described in their manuscript fully available without restriction, with rare exception (please refer to the Data Availability Statement in the manuscript PDF file). The data should be provided as part of the manuscript or its supporting information, or deposited to a public repository. For example, in addition to summary statistics, the data points behind means, medians and variance measures should be available. If there are restrictions on publicly sharing data—e.g. participant privacy or use of data from a third party—those must be specified.

Reviewer #1: Yes

Reviewer #2: Yes

5. Is the manuscript presented in an intelligible fashion and written in standard English?  PLOS ONE does not copyedit accepted manuscripts, so the language in submitted articles must be clear, correct, and unambiguous. Any typographical or grammatical errors should be corrected at revision, so please note any specific errors here.

Reviewer #1: Yes

Reviewer #2: Yes

6. Review Comments to the Author  Please use the space provided to explain your answers to the questions above. You may also include additional comments for the author, including concerns about dual publication, research ethics, or publication ethics. (Please upload your review as an attachment if it exceeds 20,000 characters)

Reviewer #1: All comments except 3 are fully addressed. Regarding the earlier comment 3. Line 181: In the models, participants “182 were censored at the time of their last test, when they did not complete the originally planned 183 five tests and had negative results in all their samples.” This statement is not clear. Were they removed completely?

 "All the participants were included in the prevalence analysis. Participants with a positive result in their first sample were excluded from the incidence analysis and were considered prevalent cases. Participants who did not complete the five follow-up visits contributed person-time from all the available tests with negative results, until the first positive result or until their last available negative result. We have included the following sentence in the text to improve clarity: “In the models, participants with negative results in all their follow-up samples were censored at the time of their last available test.”   From the response to the earlier comment above, it appears that the person time included the time till their last available negative visit if they did not complete five visits. "Participants who did not complete the five follow-up visits contributed person-time from all the available tests with negative results, until the first positive result or until their last available negative result."  “In the models, participants with negative results in all their follow-up samples were censored at the time of their last available test.” But does “censored” mean that they were removed? It is confusing if they were removed, whether only the last test was removed and also why they were removed. Or were they removed for calculation of the incident cases?

A: In this version, we eliminated the term “censored”, which is different to “removed” to avoid any confusion. Instead, we simply describe how we made the calculations. We planned five follow-up visits for blood sample donation for every participant, but only a fraction of the participants complied with the five visits. The test result of the first visit for every participant was used to define prevalent cases. Participants that resulted positive were included in the seroprevalence calculation. 

Prevalent cases were excluded from incidence estimations. All participants included in the incidence calculation contributed with follow-up time from the date of the first sample until the date of the first sample with a positive result (incident cases), or until the last date of follow up (non-cases). Please see the definition of the prevalent, incident and non-case groups, in lines 185-188. We have included the following text in the Statistical analyses section: “The person-time was calculated as follows: incident cases contributed with the time elapsed from the date of their first sample to the date of their first positive sample, and non-cases contributed with their complete follow-up time from their first to their last sample date.”

 

Additional comment 1: At the end of the study, 290 participants had a positive result in any of the antibody tests, yielding 241 an overall prevalence in the study period, adjusted by sensitivity and specificity of the antibody Line 242 test, of 33.5%. 235 positive cases (81.9%) were identified at baseline (prevalent cases), while the 243 remaining 55 (6.2%) seroconverted during the follow up period (incident cases). Among the positive cases (290) – the prevalent cases were 235/290 = 81.9%, while the incident cases would be 55/290 which is 18.9 % and not 6.2% as mentioned – is that correct?

A: Thank you for pointing out this error. We have changed the text to 18.9%, to show the percentage relative to the total positive cases. 

Additional comment 2: Line 68 - spelling error "waive" caused by the omicron variant.

A: The mistake has been corrected.

Reviewer #2: Thanks for addressing the earlier comments.  I encourage the authors to please check the manuscript for grammar and sentence structuring before final proof is sent.

A: We thank the Reviewer for the comments. The manuscript has been checked for grammar and structure with the appropriate corrections. 

7. PLOS authors have the option to publish the peer review history of their article (what does this mean?). If published, this will include your full peer review and any attached files.  Do you want your identity to be public for this peer review? For information about this choice, including consent withdrawal, please see our Privacy Policy.

Reviewer #1: No

Reviewer #2: Yes: Giridhara R Babu

---

## [Decision Letter · Decision Letter 2]

21 Feb 2022

Seroepidemiology of SARS-CoV-2 in healthcare personnel working at the largest tertiary COVID-19 referral hospitals in Mexico City.

PONE-D-21-27465R2

Dear Dr. Avila-Rios,

We’re pleased to inform you that your manuscript has been judged scientifically suitable for publication and will be formally accepted for publication once it meets all outstanding technical requirements.

Kind regards,

Leeberk Raja Inbaraj, MD

Academic Editor

PLOS ONE

Additional Editor Comments (optional):

Reviewers' comments:

Reviewer's Responses to Questions

**Comments to the Author**

1. If the authors have adequately addressed your comments raised in a previous round of review and you feel that this manuscript is now acceptable for publication, you may indicate that here to bypass the “Comments to the Author” section, enter your conflict of interest statement in the “Confidential to Editor” section, and submit your "Accept" recommendation.

Reviewer #1: All comments have been addressed

2. Is the manuscript technically sound, and do the data support the conclusions?

Reviewer #1: Yes

3. Has the statistical analysis been performed appropriately and rigorously? 

Reviewer #1: Yes

4. Have the authors made all data underlying the findings in their manuscript fully available?

Reviewer #1: Yes

5. Is the manuscript presented in an intelligible fashion and written in standard English?

Reviewer #1: Yes

6. Review Comments to the Author

Reviewer #1: The authors detailed responses to the comments is appreciated. Please do check for further transcription errors and grammatical errors.

7. PLOS authors have the option to publish the peer review history of their article (what does this mean?). If published, this will include your full peer review and any attached files.

Reviewer #1: No

---

## [Editor Report · Acceptance letter]

4 Mar 2022

PONE-D-21-27465R2 

Seroepidemiology of SARS-CoV-2 in healthcare personnel working at the largest tertiary COVID-19 referral hospitals in Mexico City. 

Dear Dr. Ávila-Ríos:

I'm pleased to inform you that your manuscript has been deemed suitable for publication in PLOS ONE. Congratulations! Your manuscript is now with our production department. 

Kind regards, 

on behalf of

Dr. Leeberk Raja Inbaraj 

Academic Editor

PLOS ONE